# The *APOE* ε4 Allele Affects Cognitive Functions Differently in Carriers of *APP* Mutations Compared to Carriers of *PSEN1* Mutations in Autosomal-Dominant Alzheimer’s Disease

**DOI:** 10.3390/genes12121954

**Published:** 2021-12-07

**Authors:** Ove Almkvist, Caroline Graff

**Affiliations:** 1Division of Clinical Geriatrics, Department of Neurobiology Care Sciences and Society, Karolinska Institutet, SE-14157 Stockholm, Sweden; 2Theme Aging, Karolinska University Hospital, SE-14157 Stockholm, Sweden; caroline.graff@ki.se; 3Department of Psychology, Stockholm University, SE-10691 Stockholm, Sweden; 4Division of Neurogeriatrics, Department of Neurobiology Care Sciences and Society, Karolinska Institutet, SE-14157 Stockholm, Sweden

**Keywords:** *APOE*, autosomal-dominant Alzheimer’s disease, *APP*, *PSEN1*, cognition, epistasis

## Abstract

Mounting evidence shows that the *APOE* ε4 allele interferes with cognition in sporadic Alzheimer’s disease. Less is known about *APOE* in autosomal-dominant Alzheimer’s disease (adAD). The present study explored the effects on cognition associated with the gene–gene interactions between the *APOE* gene and the *APP* and *PSEN1* genes in adAD. This study includes mutation carriers (MC) and non-carriers (NC) from adAD families with mutations in *APP* (*n* = 28 and *n* = 25; MC and NC, respectively) and *PSEN1* (*n* = 12 and *n* = 15; MC and NC, respectively) that represent the complete spectrum of disease: AD dementia (*n* = 8) and mild cognitive impairment (MCI, *n* = 15 and presymptomatic AD, *n* = 17). NC represented unimpaired normal aging. There was no significant difference in the distribution of *APOE* ε4 (absence vs. presence) between the *APP* vs. *PSEN1* adAD genes and mutation status (MC vs. NC). However, episodic memory was significantly affected by the interaction between *APOE* and the *APP* vs. *PSEN1* genes in MC. This was explained by favorable performance in the absence of *APOE* ε4 in *PSEN1* compared to *APP* MC. Similar trends were seen in other cognitive functions. No significant associations between *APOE* ε4 and cognitive performance were obtained in NC. In conclusion, cognitive effects of *APOE*–adAD gene interaction were differentiated between the *PSEN1* and *APP* mutation carriers, indicating epistasis.

## 1. Introduction

Alzheimer’s disease (AD) is defined at autopsy by extracellular deposits of β-amyloid accumulated in plaques and neurofibrillary tangles of intracellular phosphorylated tau [1]. According to empirical evidence, the *APOE* ε4 allele has a deleterious effect on disease onset in sporadic Alzheimer’s disease (sAD) that is related to the number of *APOE* ε4 alleles [2]. *APOE* ε4 influences lipid metabolism in the brain, β-amyloid and tau processing, synaptogenesis, glucose metabolism, mitochondrial function, vascular integrity, and neuroimmune modulation [3,4]. In this way, sAD as well as normal aging is affected by various *APOE*-related functions. However, the precise mechanisms for these effects are far from known.

Less is known about the relationship between *APOE* and possible effects in autosomal-dominant AD (adAD). In recent studies of adAD, the *APP*, *PSEN1* and *PSEN2* genes have been combined into one group of mutation carriers. The impact of *APOE* ε4 on β-amyloid burden has not been clear, making a conclusion impossible in adAD [5,6]. Years to estimated clinical onset (YECO) was driving changes across time. The YECO–*APOE* ε4 interaction did not add any power to the time-related cognitive changes in adAD. In a cohort of the specific Colombian *PSEN1* mutation, no significant effect was found in association with *APOE* ε4 [7].

The combination of mutation carriers from various genes into one combined group may be problematic, because adAD genes and mutations may be associated with different phenotypes of cognition in *APP* and *PSEN1* genes [7,8,9,10,11].

The aim of the present study was to investigate the possible interaction between the *APOE* gene and the *APP* and *PSEN1* genes in adAD as observed on various cognitive functions.

## 2. Methods

### 2.1. Participants

Adult mutation carriers (MC, *n* = 40) and non-carriers (NC, *n* = 40) from six adAD families harboring a mutation in the *APP* or *PSEN1* genes were invited to the Memory Clinic, Karolinska University Hospital Huddinge, Sweden, to participate in research. Three families carried an *APP* mutation (Swedish, p.KM670/671NL; Arctic, p.E693G and London, p.V717I) and three families carried a *PSEN1* mutation (p.I143T, p.M146V, and p.H163Y). The demographic characteristics (age, gender, years of education, years to expected clinical onset of disease and *APOE* ε4 percentage) of MC and NC are presented in Table 1.

Analyses of the comparability of MC and NC groups from *APP* and *PSEN1* families showed that *APP* individuals were significantly older than *PSEN1* individuals (F = 25.52, df 1/76, *p* < 0.001, η^2^ = 0.25), while advancement of disease as years to estimated clinical onset (YECO; see below) was not significant as well as the effects of gender, education and the *APOE* ε4 allele (present vs. absent). There were no significant differences between MC and NC in background characteristics or significant interaction effects, except for years of education (F = 9.34, df 1/53, *p* = 0.004, η^2^ = 0.15). These results support that the groups of MC and NC in *APP* and *PSEN1* families are roughly comparable in background characteristics.

### 2.2. Examination and Diagnosis

The standardized clinical examination included somatic, neurological, and psychiatric status, often an interview with a close informant, cognitive screening (MMSE) [12], cognitive assessment, Magnetic imaging, electric encephalography, and biochemical assessments of urine, blood, Cerebrospinal fluid markers (Abeta, total tau and phosphorylated tau), and fibroblast. Studies on non-cognitive outcomes have been reported elsewhere [13,14].

The clinical diagnosis was decided by a consensus meeting of medical professionals and based on all available reports, excluding information on the mutation status. At baseline, eight carriers (6 *APP* and 2 *PSEN1*) were diagnosed with dementia [15] and as having AD [16], 15 were diagnosed with mild cognitive impairment (MCI, 8 *APP* and 7 *PSEN1*) [17] and 17 (14 *APP* and 3 *PSEN1*) were asymptomatic. No NC was diagnosed with AD, MCI, or any other disease affecting the brain, neither at baseline nor at any follow-up examination.

### 2.3. APOE Genotyping

*APOE* genotyping was performed for SNPs rs7412 and rs429358 using TaqMan^®^, SNP Genotyping Assays (ABI, Foster City, CA, USA) according to manufacturer’s protocol. The amplified products were run on the 7500 fast Real-Time PCR Systems (ABI, Foster City, CA, USA).

### 2.4. Years to Estimated Clinical Onset (YECO)

The years to estimated clinical onset of disease was calculated as the participant’s present age minus the family-specific age of clinical onset in correspondence with previous research [18,19]. This measure is invariant within families, variable between families and significantly associated with the observed age of onset and parental age of onset [18,19].

### 2.5. Cognitive Assessment

The cognitive assessment focused on tests varying in sensitivity for the development of AD [17,20]. The earliest change typically occurs in episodic memory, followed by executive function and visuospatial ability, and later by attention/processing speed and verbal ability. In the present study, the tests of Rey Auditory Verbal Learning (RAVL, episodic memory), Digit Symbol (executive function), Block Design (visuospatial ability), the Trailmaking A test (TMT A, attention) and Similarities (verbal ability) were used as outcome measures [18]. Raw test scores were standardized into z-scores based on data from normal individuals [21].

### 2.6. Ethical Approvement

All participants were given written and oral information about the risk to inherit AD before this study began. All participants provided written informed consent to participate in this study. This study was approved by the Regional Ethics Committee in Stockholm (2006/901-31/3). This study was performed according to the declaration of Helsinki and subsequent revisions.

### 2.7. Statistical Analyses

A two-way ANOVA analyzed the distribution of *APOE* ε4 (present/absent) and demographic characteristics as related to mutation status (MC vs. NC) and adAD genes (*APP* vs. *PSEN1*). Five two-way ANOVAs, one for each cognitive test, analyzed the effect of the *APOE* gene, the adAD genes (*APP* vs. *PSEN1*) and their interaction separately in mutation carriers and non-carriers. Two-way ANCOVAs were used to analyze the effect on tests related to mutation (MC/NC) and gene (*APP*/*PSEN1*) with YECO and education as covariates.

## 3. Results

The cognitive test results across the five domains for *APP* and *PSEN1* MC and NC are presented in Table 2. In NC, the results scatter around the mean performance for cognitively unimpaired individuals as expected. In contrast, the test results for MC vary from normal to clearly impaired as seen in episodic memory in *APP* MC, although the majority are in the preclinical stage of disease. The cognitive trajectory for episodic memory of these cross-sectional test results across time (YECO) is presented in Figure 1. The *APP* MC seem to start the decline decades before the clinical diagnosis is possible to verify. The *PSEN1* MC seem to keep a normal performance level until approximately 10 years ahead of the clinical diagnosis; then the decline is steep.

The possible effect of mutation (MC/NC) and gene (*APP*/*PSEN1*) together with covariates (YECO and education) on test results are presented in Table 2. Mutation status was significant in four tests, gene in one test and the interaction was not significant in any test. YECO was significantly influencing test results in all tests and years of education was significant in four tests.

In MC, there was a significant effect of the *APOE*–adAD gene interaction on episodic memory (the RAVL learning test) (F = 6.66, df = 1/33, *p* = 0.014, η^2^ = 0.17), see Figure 2. The graph shows a typical crossed pattern seen in statistical interaction: The *APP* MC performed poorer in episodic memory (*p* > 0.1), when *APOE* ε4 was absent (M ± SD: −1.62 ± 1.33) compared to present (M ± SD: −0.93 ± 1.38), while the opposite was seen in *PSEN1* MC, namely significantly (*p* = 0.04) better performance when *APOE* ε4 was absent (M ± SD: 0.47 ± 0.71) compared to present (M ± SD: −1.25 ± 1.71).

Among the other four cognitive tests, there was a trend of significant interaction effect in visuospatial ability (Block Design) (F = 3.46, df = 1/36, *p* = 0.071, η^2^ = 0.09), and in executive function (Digit Symbol) (F = 3.03, df = 1/20, *p* = 0.10, η^2^ = 0.13), while the *APOE*–adAD gene interaction was not significant in verbal ability (Similarities) and attention/processing speed (p’s > 0.1). The pattern of results across the four cognitive tests demonstrated the same pattern as seen in episodic memory, i.e., a favorable performance for *PSEN1* MC when the *APOE* ε4 was absent compared to present. For *APP* MC, the results were better when the ε4 allele was present compared to absent. In executive function (Digit Symbol), this effect was close to significant (*p* = 0.053), better in ε4 carriers (M ± SD: −0.11 ± 1.34) than in ε4 non-carriers (M ± SD: −1.26 ± 1.34). There were no significant interaction effects in NC. There were no significant main effects of *APOE* or adAD gene in any cognitive test (all *p*’s > 0.1).

## 4. Discussion

This study investigated the possible interaction effect on cognitive functions between *APOE* and the *APP* and *PSEN1* genes in adAD in a sample of MC and NC from six families harboring an adAD gene. This study covered the whole range of cognitive impairment from the presymptomatic stage across mild cognitive impairment and finally AD dementia, i.e., more than 40 years (30 years before and 10 years after the expected onset). The *APP* and *PSEN1* MC were comparable in background factors.

The main finding was that the *APOE*–adAD gene interaction was significant in episodic memory (the RAVL learning test) that accounted for 17% of the total test variance. Furthermore, there was a trend to significant interaction effect in visuospatial ability that accounted for 9% of the variance (the Block Design test) and in executive function that accounted for 13% of the variance (the Digit Symbol test). The statical interaction was observed as a favorable and significant effect in *PSEN1* MC when *APOE* ε4 was absent compared to a favorable effect on cognition in *APP* MC when *APOE* ε4 was present (although not significant), i.e., opposite interaction effects.

In addition, a similar pattern of segregated ε4 effects (absence vs. presence) for *PSEN1* vs. *APP* MC were obtained in the other cognitive tests. The *APOE*–adAD gene interaction effect was significant and favorable in *APP* MC on executive function (the Digit Symbol test), when *APOE* ε4 was present, while the interaction effect was positive (although not significant) in *PSEN1* MC, when *APOE* ε4 was absent, i.e., opposite.

The interaction between the *APOE* and adAD genes have not been reported previously to our knowledge. The finding was based on a relatively small convenient sample of participants carrying a mutation in the *APP* or *PSEN1* genes in adAD. The *PSEN2* gene was not represented and several mutations in the *APP* and *PSEN1* adAD genes were not investigated in the present study. These limitations make it necessary to await larger studies until a more reliable conclusion can be reached.

That said, the gene–gene interactions observed in the present study point to the possible complexity of function involved in the *APOE* gene. This complexity has recently been dealt with in recent publications. It seems clear that *APOE* is a significant factor in sAD [5,6] and that the *APOE* ε4 allele has gain of function effect in the earliest ontogenesis [22], in aging [23], in environments differing in infectious burden in Amazonas versus Western societies [24], and in demanding cognitive tasks [23] compared overlearned cognitive tasks [25,26]. Assuming that the findings are valid, they illustrate that the *APOE* gene is involved in both gene–gene and gene-environment interactions seen both within individuals across tasks and between individuals. This effect has been suggested to be a response to varying degree of neural stress [25].

The lack of knowledge about *APOE* functions may depend on the fact that many mechanisms are involved in *APOE* functioning, not only those related to lipid metabolism, β-amyloid and tau processes, but also synaptogenesis, glucose metabolism, mitochondrial function, vascular integrity, and neuroimmune modulation [3,4]. These various functions of *APOE* have led to contrasting outcomes like resistance to toxicity and loss of neuroprotection, summarized as the phenomena of pleiotropi.

Furthermore, we have previously shown that *APP* processing varies between *APP* and *PSEN1* mutations as well as between different mutations in the same gene [13]. Thus, it is possible that the *APOE*-adAD gene interaction in parts reflects differences in the interaction caused by the different *APP*-processing products generated via the amyloidogenic as well as non-amyloidogenic pathways.

In conclusion, an interaction was observed between the *APOE* ε4 allele and the *APP* and *PSEN1* genes in adAD. The effect of *APOE* ε4 on cognition was favorable in *PSEN1* MC without an *APOE* ε4 allele and favorable in *APP* MC in the presence of an *APOE* ε4 allele. These results are an indication of epistasis.

## Figures and Tables

**Figure 1 genes-12-01954-f001:**
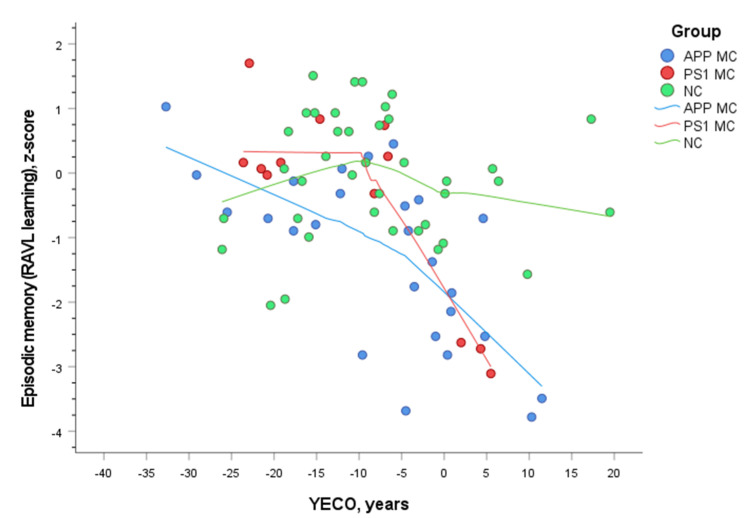
The trajectories for *APP* and *PSEN1* MC as well as all NC across YECO using cross-sectional data.

**Figure 2 genes-12-01954-f002:**
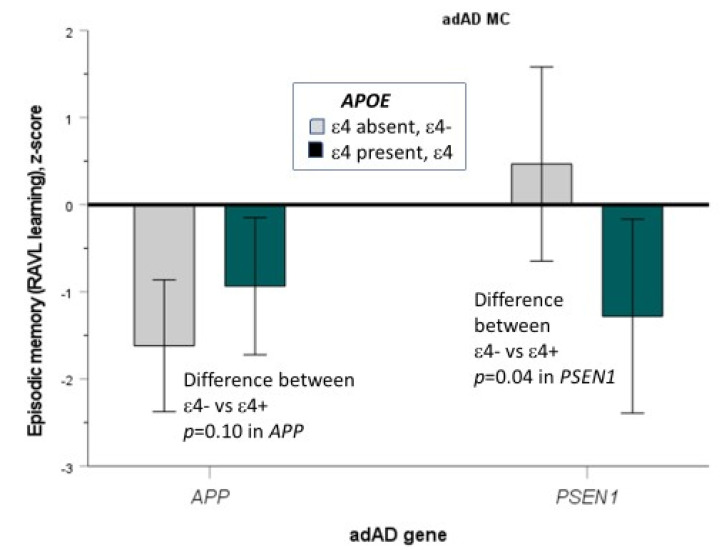
A graph showing mean (95% CI) episodic memory performance (RAVL learning) between *APOE* ε4positive and ε4 negative mutation carriers of *APP* and *PSEN1* adAD genes.

**Table 1 genes-12-01954-t001:** Demographic characteristics (age, gender, YECO = years to estimated clinical onset, years of education, % *APOE* ε4) of adAD mutation carriers (MC) and non-carriers (NC) from *APP* and *PSEN1* families. There were 12 MC with *APP* Swedish, 15 with *APP* Arctic, 1 with *APP* London, 3 with *PSEN1* I143T or M146V and 9 with *PSEN1* H163Y mutation. In addition, *p*-values are reported using two-way (Mutation and Gene) ANOVAs with each background factor as dependent variable.

	APP		PSEN1		*p*		
	MC	NC	MC	NC	M	G	MxG
N (% females)	28 (36)	25 (44)	12 (33)	15 (47)	ns	ns	ns
Age, years (Mean ± SD)	47.9 ± 11.2	47.4 ± 9.9	36.2 ± 9.3	33.8 ± 11.1	ns	***	ns
YECO (Mean ± SD)	−7.1 ± 11.0	−7.5 ± 9.7	−11.1 ± 10.9	−8.2 ± 11.8	ns	ns	ns
Education, years (Mean ± SD)	10.9 ± 2.5	10.8 ± 2.9	11.8 ± 3.0	10.6 ± 1.5	ns	ns	ns
*APOE* e4, %	46	28	50	40	ns	ns	ns

Note. M = (MC/NC), G = (*APP*/*PSEN1*), MxG = interaction, ns = not significant, and *** = *p* < 0.001.

**Table 2 genes-12-01954-t002:** Cognitive test results (Mean ± SD) in z-score based on normal individuals in mutation carriers (MC) and non-carriers (NC) from *APP* and *PSEN1* families. ANCOVA *p*-values are reported as well as related to mutation status, gene, mutation–gene interaction and the effects of covariates (YECO and years of education).

Domain/Test	APP		PSEN1		*p*				
	MC	NC	MC	NC	M	G	MxG	Y	E
Verbal/Similarities	−1.18 ± 1.56	−0.54 ± 0.92	−0.28 ± 1.67	−0.88 ± 0.98	ns	ns	ns	*	**
Visuospatial/Block Design	−0.71 ± 1.79	+0.05 ± 1.54	+0.26 ± 2.33	+0.73 ± 1.03	*	ns	ns	***	***
Episodic memory/RAVL	−1.29 ± 1.37	−0.12 ± 0.83	−0.41 ± 1.55	+0.03 ± 1.13	***	ns	ns	***	**
Executive/Digit Symbol	−0.73 ± 1.58	+0.28 ± 0.99	+0.35 ± 1.70	+0.63 ± 1.08	**	*	ns	***	***
Attention/TMT A	−1.14 ± 3.00	+0.56 ± 2.08	+0.05 ± 2.25	+0.65 ± 0.78	*	ns	ns	**	ns

Note. Note. M = (MC/NC), G = (*APP*/*PSEN1*), MxG = interaction, Y = YECO, E = education, * = *p* < 0.05, ** = *p* < 0.01, *** = *p* < 0.001, and ns = not significant.

## Data Availability

Data are available upon reasonable request to the corresponding author or the principal investigator (Caroline Graff).

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
