# Peer review of "The APOE ε4 Allele Affects Cognitive Functions Differently in Carriers of APP Mutations Compared to Carriers of PSEN1 Mutations in Autosomal-Dominant Alzheimer’s Disease"

_genes, 2021, doi:10.3390/genes12121954_

Round 1

Reviewer 1 Report

Author conducted a study to investigate the effects on cognition associated with the gene-gene interactions between the APOE gene and the APP and PSEN1 genes in autosomal-dominant Alzheimer’s disease (adAD). They found an interaction between the APOE and the APP and PSEN1 genes in adAD. They identified that the APP MC showed unfavorable performance in episodic memory in absence of APOE e4 allele. However, for PSEN1 MC there was a better performance in absence of APOE e4 allele, representing epistasis. 

However, some comments must be considered:

 1-Due to small sample size, the result of their study could be affected and might be changed with higher number of samples and different mutation types.

2-Were mutations in main protein domains or other domains?

3- Since different mutation types are involved in adAD forms, it would better to consider a wide range of mutation types in different protein domains of APP and PSEN1.

4- It is also suggested to look for this effect in terms of correlation between mutation types in APP and PSEN1 and APOE e4 allele. Author should check which mutation in these genes was related to more unfavorable cognitive performance in absence of APOE 4 in APP MC vice versa.

Author Response

Responses to the reviewers re: reviewer 1

Comment 1

Due to small sample size, the result of their study could be affected and might be changed with higher number of samples and different mutation types.

Our response

We agree with the reviewer that the sample size is small and that the number of different mutations is small. This fact is not possible to change at present. Furthermore, the number of individuals with each mutation is also small. The number of participants within each mutation-type has been added to the manuscript, see Table 1. This fact hampers the possibility to analyse mutation-specific effects, see also the response to point 2. Therefore, the study is limited to the study of gene-related effects and any mutation-type effects requires a larger sample size. See also the response regarding point 4.

Comment 2

Were mutations in main protein domains or other domains?

Our response

To address this question, we have added information in the method section regarding participants with the different mutations in APP and PSEN1 genes: “Three families carried an APP mutation (Swedish, p.KM670/671NL; Arctic, p.E693G and London, p.V717I) and three families carried a PSEN1 mutation (p.I143T, p.M146V, and p.H163Y).”

We have also added the number of MC per mutation, see Table 1. Due to the small number of MC in the PSEN1 I143T family, we pooled this family with the PSEN1 M146V family to avoid unintentional disclosure of mutation status for family members.

Finally, we introduced a new paragraph to the discussion in which we speculate that APOE-by-adAD gene interactions may be caused, in part, by differences in APP-processing secondary to the type of mutation. This idea is supported by findings showing variation in APP-processing in APP and PSEN1 genes and variation in APP-processing in different mutation within the same gene (Thordardottir et al., 2017). The reference has now been added to the ms.

Comment 3

Since different mutation types are involved in adAD forms, it would better to consider a wide range of mutation types in different protein domains of APP and PSEN1.

Our response

We agree with the reviewer on this point. The number of MC and NC is small both in the APP (28 vs 25) and PSEN1 genes (12 vs 15) as now presented in Table 1. This is further illustrated when participants are divided into APOE e4+ and e4- cases. In future studies it may be possible to enlarge the number of participants by international collaboration.

Comment 4

It is also suggested to look for this effect in terms of correlation between mutation types in APP and PSEN1 and APOE e4 allele. Author should check which mutation in these genes was related to more unfavorable cognitive performance in absence of APOE e4 in APP MC vice versa.

Our response

Now, we have analysed the data within specific mutations. There are three mutations in which the number of participants are >15 (APP Swedish, APP Arctic, and the PSEN1 H163Y mutations), but when these groups are divided into APOE e4+ MC and e4- MC, the sample size is increasingly limited (4 vs 10 MC, 6 vs 8 MC and 4 vs 2 MC in e4+ vs e4- subgroups, respectively) making it hard find consistent patterns of results. With the present number of MC in subgroups of adAD genes (MC vs NC) and APOE e4 (presence vs absence) some trends, far from significant, were observed. We prefer not to report unreliable results on this point.

Finally, we would like to point out that it was not possible to use a dose-response analysis to investigate the APOE e4 effect as related to a mechanism in this study because the number APOE e4 with two e4 alleles was only three individuals, one in each of three subgroups (APP NC, PSEN1 NC and APP MC).

Reviewer 2 Report

Authors investigated the APOE-e4 allele effects on cognition (Table 2, and
Figures 1 and 2) in adAD families (6 families, including 80 subjects, in total;
Table 1) carrying APP or PSEN1 rare missense mutations: Swedish, Arctic,
and London mutations for APP, and I143T, M146V, and H163Y mutations for
PSEN1. Through a series of analyses, they found that episodic memory
(RAVL learning test) was significantly affected by the interaction between
APOE and the APP vs PSEN1 genes in MC. Basically, I agree with authors
idea although sample size seems to be not enough to finally conclude the
significant interaction, as authors mentioned in Discussion (lines 201 and
202). However, unfortunately, no P-values are given in Tables 1 and 2, itself.
It is important for readers to easily understand which items/tests analyzed
are statistically significant or not at a glance. Thus, as one of minor revision
points, I would like to suggest the attachment of P-values in each Table. Also,
it is helpful for readers if graphs such as box/violin plots concerning cognitive
test results, given in Table 2, are kindly presented. There are some
additional very minor revision points in the main body of the manuscript,
depicted below.

Very minor revision points:
1) Line 15: ... APP (n=28 and n=25) ... ---> ... APP (n=28 in MC, and n=25 in
NC) ...? Could you separately describe number of subjects used in the
study?
2) Line 16: ... PSEN1 (n=12 and n=15) ---> ... PSEN1 (n=12 in MC, and
n=15 in NC) ...? It is same above.
3) Line 17: ... Mild Cognitive Impairment (MCI, n=15 and presymptomatic
AD (n=17). ---> ... Mild Cognitive Impairment (MCI, n=15), and
presymptomatic AD (n=17)? Is it right?
4) Line 23: No space between ... NC. and In conclusion, ....
5) Lines 71, 72, and 73 in Table 1: Mean, or Median + Standard deviation
(SD)? Could you properly mention in the table legend?
6) Lines 88, and 89: MRI, EEG, CSF? No abbreviation information.
7) Lines 89, and 90: Concerning the sentence of “Studies on non-cognitive
outcomes have been reported elsewhere., no paper is cited. If possible, it

is helpful for readers to cite one of representative papers.
8) Line 113: RAVL? No abbreviation information.
9) Line 114: TMT? No abbreviation information.
10) Line 140 in Table 2: M? SD? No abbreviation information.
11) Line 141 in Table 2: PSEN1 is not italic.
12) Line 143 in Table 2: PSEN1 is not italic.
13) Line 193: The APOE-by-asAD... ---> ... The APOE-by-adAD...

Author Response

Responses to the reviewers re: reviewer 2

Comment

Basically, I agree with authors’ ideal though sample size seems to be not enough to finally conclude the significant interaction, as authors mentioned in Discussion (lines 201 and 202).

Our response

We agree with the reviewer that the sample size is small, see our response on point 1 above.

Comment

However, unfortunately, no P-values are given in Tables 1 and 2, itself. It is important for readers to easily understand which items/tests analyzed are statistically significant or not at a glance. Thus, as one of minor revision points, I would like to suggest the attachment of P-values in each Table. Also, it is helpful for readers if graphs such as box/violin plots concerning cognitive test results, given in Table 2, are kindly presented.

Our response

Now we have added p-values to tables 1 and 2 as suggested by the reviewer. We have also added p-values to Figure 2 to clarify the results of Table 2 as suggested by the reviewer.

Comments regarding minor revision points in the main body of the ms.

1) Line 15: ... APP (n=28 and n=25) ... ---> ... APP (n=28 in MC, and n=25 in NC) ...? Could you separately describe number of subjects used in the study?

2) Line 16: ... PSEN1 (n=12 and n=15) ---> ... PSEN1 (n=12 in MC, and n=15 in NC) ...? It is same above.

3) Line 17: ... Mild Cognitive Impairment (MCI, n=15 and presymptomatic AD (n=17). ---> ... Mild Cognitive Impairment (MCI, n=15), and presymptomatic AD (n=17)? Is it right?

4) Line 23: No space between “... NC.” and “In conclusion, ...”.

5) Lines 71, 72, and 73 in Table 1: Mean, or Median + Standard deviation (SD)? Could you properly mention in the table legend?

6) Lines 88, and 89: MRI, EEG, CSF? No abbreviation information.

7) Lines 89, and 90: Concerning the sentence of “Studies on non-cognitive outcomes have been reported elsewhere.”, no paper is cited. If possible, itis helpful for readers to cite one of representative papers.

8) Line 113: RAVL? No abbreviation information.

9) Line 114: TMT? No abbreviation information.

10) Line 140 in Table 2: M? SD? No abbreviation information.

11) Line 141 in Table 2: “PSEN1” is not italic.

12) Line 143 in Table 2: “PSEN1” is not italic.

13) Line 193: The APOE-by-asAD... ---> ... The APOE-by-adAD...

Our response

We thank the reviewer for careful reading and pointing out all our errors. Now, errors have been corrected accordingly. The noticed abbreviations have been spelt out in the ms (see points 6, 8-10).

Round 2

Reviewer 1 Report

While there are some limitations in this study, authors showed an interesting study that cognitive effects of APOE-by-adAD interaction were differentiated between the PSEN1 and APP genes.